# Prevalence of previously diagnosed diabetes and glycemic control strategies in Mexican adults: ENSANUT-2016

Ismael Campos-Nonato[1]◉, María Ramírez-Villalobos[2]◉*, Alejandra Flores-Coria[1]◉, Andrys Valdez[1]◉, Eric Monterrubio-Flores[1]◉

1 Centro de Investigación en Nutrición y Salud, Instituto Nacional de Salud Pública, México City, Mexico,
2 Centro de Investigación en Sistemas, Instituto Nacional de Salud Pública, México Cty, Mexico

◉ These authors contributed equally to this work.
* dolores.ramirez@insp.mx

**Data Availability Statement:** The data underlying this study were generated by third parties (National Institute of Public Health), and are freely available.

## Abstract

### Objectives

To describe the prevalence of previously diagnosed diabetes among Mexican adults, to characterize the associated risk factors, and to describe which glycemic control strategies are the most used.

### Methods

We analyzed data from 8,631 adults aged ≥20 years who participated in the ENSANUT-2016 and from whom we gathered data about previously diagnosed diabetes, risk factors, glycemic control strategies, and measures to prevent complications.

### Results

The prevalence of previously diagnosed diabetes in Mexican adults was 9.4% (10.3% in women and 8.4% in men). The adjusted OR for having diabetes was higher in adults aged ≥60 years (OR = 11.0 in women and OR = 30.7 in men) than in adults aged 20–39 years (OR = 1.0). The adjusted OR for having diabetes was higher in overweight men (OR = 1.7) than in men with normal BMI (OR = 1.0). A total of 30.5% of adults with diabetes did not report any control strategies, 44.9% measured their venous blood glucose, and 15.2% used the HbA1C as an indicator of glycemic control. Only 46.4% of them reported preventive measures.

### Discussion

Diabetes is a common disease among Mexican adults. Being older or overweight are risk factors for an adult to be diagnosed with diabetes. Most adults with diabetes evaluate their glycemic control but only half practice preventive measures.

The results can be replicated using the same methodology described in our manuscript. The information from the survey is available at the following link: https://ensanut.insp.mx/encuestas/ensanut2016/descargas.php. The authors used the data from the household questionnaire to analyze socioeconomic and demographic variables. We also use information from the database of adults over 20 years of age (anthropometry, physical activity and section of chronic diseases such as diabetes, hypertension, and others). The authors did not have special access privileges.

**Competing interests:** The authors have declared that no competing interests exist.

## Introduction

Diabetes is a chronic disease that during its first stages produces no symptoms and that when not adequately treated causes complications such as retinopathy, nephropathy, heart attack and premature death [1].

There are non-modifiable risk factors, such as genetics; but others are indeed modifiable such as obesity [2], diet [3], screen time [4], sleep quality [5], and tobacco smoking. [6] Prevention and management of modifiable risk factors for diabetes type 2 could delay or prevent the occurrence of complications and improve its control.

During 2014, the prevalence of diabetes type 2 among adults was 8.5% [7] worldwide, and in Mexico, according to the national health survey 2012, the prevalence was 9.2%. [8]

Due to the multicausality and chronicity of diabetes, people living with this disease need lifestyle interventions to prevent or minimize its progression, continuous medical attention to assess and control glycemia according to its clinical response, and preventive measures to avoid and delay the occurrence of complications. [9, 10].

Identifying the risk factors associated with the diagnosis and control of diabetes in a population contributes to showing the areas to which the strategies for screening, hyperglycemia treatment and prevention of potential complications should be directed [5]. In Mexico, there is no recent evidence that characterizes at a national level the risk factors associated with the diagnosis of diabetes and to its control. Therefore, the objective of our study is to describe the prevalence of previously diagnosed type 2 diabetes in adults that live in Mexico, to characterize the associated risk factors, and to describe the most used glycemic control strategies.

## Methodology

The National Health and Nutrition Survey 2016 (ENSANUT-2016, Encuesta Nacional de Salud y Nutricion 2016) followed a cross-sectional, probabilistic, regional representative, and by area of residence (urban ≥2,500 inhabitants and rural <2,500 inhabitants) design. A total of 9,406 adults were selected achieved a 91.7% response rate. The detailed description of the sampling procedures, survey methodology, regionalization (North, Center, Mexico City, and South), and the socioeconomic status (SES) configuration (low, medium, and high) has already been published elsewhere. [11]

### Participants

In the analysis 8,631 adults that had full information about the previously diagnosed diabetes, risk factors, treatment, screening of the disease, associated complications, and complications preventive measures were included.

### Diabetes medical diagnosis

The prevalence of previously diagnosed diabetes was determined based on the question: "Has a doctor ever told that you have diabetes or high blood sugar?". We considered that there was a prior diagnosis of diabetes when the participants answered "Yes".

**Associated chronic diseases.** We considered that a participant had any of the following diseases: high blood pressure, kidney failure, cerebrovascular disease, acute myocardial infarction or angina, when the participant self-reported that a medical doctor had diagnosed that pathology throughout his life.

Measurements for weight, size, and waist circumference were collected by trained and standardized staff using internationally accepted protocols. [12,13]. Weigh was measured using an electronic scale with an accuracy of 100 g and height was measured using a stadiometer with

an accuracy of 2 mm. The World Health Organization (WHO) criteria was used to classify body mass index (BMI) into three categories: normal BMI (18.5–24.9 kg/m$^2$), overweight (25.0–29.9 kg/m$^2$), obesity ($\geq$30.0 kg/m$^2$). [14]

**Physical activity and screen time.** To determine the level of physical activity (PA), the short version of the International Physical Activity Questionnaire (IPAQ) was used. [15] To categorize the level of moderate to vigorous PA performed during the last seven days, we used the WHO classification: inactive <150 minutes, moderately active 150–299, and active >300. [16] To categorize the screen time, the minutes per week of tv watching, videogaming, and computer use were counted and then divided into three groups: $\leq$840, 840–1680, and >1680. [17]

**Dietary diversity (DD).** A questionnaire on frequency of food intake (FFQ) was administered seven days before the interview. The FFQ included 140 foods that were divided in 22 food groups. [18] In the DD index, the number of food groups (2–22 groups) was weighed against the number of days they were consumed during the week. The DD score was 2–154 points. Afterwards, the score was classified into quartiles.

Sleep quality was measured with the question "how would you rate the quality of your sleep regularly?" the possible answers were: good or very good and bad or very bad. The strategies to assess glycemic control during the last year and the measures to prevent complications associated were self-reported only by participants previously diagnosed with type 2 diabetes.

## Statistical analysis

We estimated the prevalence and the confidence interval at 95% (95% CI) of previously diagnosed diabetes, strategies to assess glycemic control during the last year and measures to prevent complications associated with diabetes, categorizing by variable of interest. We also calculated the odds rate (OR) for having diabetes, adjusting for sociodemographic, anthropometric, and clinical variables. A $p<0.05$ value was considered statistically significant. All analyses were conducted using the SVY module for complex samples of the statistical software STATA, version 14 (College Station, TX, USA).

## Ethical considerations

All participants signed an informed consent approved by the Institutional Review Board of the National Institute of Public Health in Mexico. This study is based on an analysis of databases, the original protocol has the approvals of the ethical and research commissions of the National Institute of Public Health, with Commission number 1401, registration with Conbioetics: 17 CEI00120130424, registration with COFEPRIS CEI 17 007 36

## Results

The prevalence of previously diagnosed diabetes was 9.4% in Mexican adults (10.3% in women and 8.4% in men). When categorizing by age groups, we observed that among 20–39 year-old-adults, the prevalence of diabetes was 3.8 times higher in women than in men (Table 1). Women with primary or less education were also more likely to be diagnosed with diabetes (18.8%; CI95% 14.9–23.4) than their male counterparts (12.5%; CI95% 10.4–14.8).

Table 2 shows that among women as well as among men, the adjusted OR for having diabetes was significantly higher (p<0.01) in the $\geq$60 year-old-group (OR = 11.0 in women and OR = 30.7 in men) than in the 20–39 year-old-group (OR = 1.0). In overweight men, the OR for having diabetes was higher (1.7 CI 95% 1.1–3.0) than in normal BMI men; and in hypertense men, (4.2 CI 95% 2.5–6.9), the OR was higher than in men with no hypertension. In women, having a cerebrovascular disease, high blood pressure, acute myocardial infarction, or

**Table 1. Prevalence of previously diagnosed diabetes in ≥ 20 year-old Mexican adults.** ENSANUT 2016-Mexico[*].

| | Previously Diagnosed Diabetes | | | | | | | | |
|---|---|---|---|---|---|---|---|---|---|
| | Total | | | Women | | | Men | | |
| | n | n thousands | Prevalence (95% CI) | n | n thousands | Prevalence (95% CI) | n | n thousands | Prevalence (95% CI) |
| **National** | 972 | 6,464.8 | 9.4 (8.2–10.8) | 664 | 3,771.6 | 10.3 (8.7–12.4) | 308 | 2,693.2 | 8.4 (7.0–10.1) |
| **Age (years)** | | | | | | | | | |
| 20–39 | 69 | 521.9 | 1.5 (1.1–2.1) | 54 | 428.8 | 2.3 (1.5–3.4) | 15 | 93.2 | 0.60 (0.3–1.1) |
| 40–59 | 415 | 2,742.2 | 12.3 (10.5–14.5) | 290 | 1,454.7 | 11.9 (9.8–14.5) | 125 | 1,287.5 | 12.8 (10.1–16.3) |
| 60 and more | 488 | 3,200.6 | 27.4 (23.2–31.9) | 320 | 1,888.1 | 30.6 (24.4–37.6) | 168 | 1,312.5 | 24.1 (19.1–28.2) |
| **Socioeconomic tertile** | | | | | | | | | |
| Low | 448 | 2,374.6 | 10.3 (8.1–12.9) | 319 | 1,575.1 | 12.8 (9.2–17.7) | 129 | 799.5 | 7.4 (5.7–9.4) |
| Medium | 331 | 2,288.1 | 10.1 (8.3–12.2) | 221 | 1,229.1 | 10.0 (8.0–12.1) | 110 | 1,059.0 | 10.3 (7.3–14.1) |
| High | 193 | 1,802.1 | 7.9 (6.3–10.1) | 124 | 967.3 | 8.1 (5.9–10.9) | 69 | 834.7 | 7.8 (5.2–11.5) |
| **Education level** | | | | | | | | | |
| Primary or less | 655 | 3,756.2 | 15.9 (13.5–18.6) | 457 | 2,411.8 | 18.8 (14.9–23.4) | 198 | 1,344.4 | 12.5 (10.4–14.8) |
| Secondary or high school | 246 | 2,068.1 | 6.7 (5.4–8.3) | 168 | 1,048.5 | 6.5 (5.1–8.3) | 78 | 1,019.6 | 7.1 (4.7–10.1) |
| Bachelor's degree | 71 | 640.5 | 4.5 (3.3–6.3) | 39 | 311.3 | 4.1 (2.5–6.4) | 32 | 329.2 | 5.2 (3.2–8.1) |
| **Area of residence** | | | | | | | | | |
| Rural | 422 | 1,479.7 | 9.3 (7.8–10.9) | 292 | 783.2 | 9.4 (8.1–11.1) | 130 | 696.5 | 9.1 (6.8–11.9) |
| Urban | 550 | 4,985.1 | 9.5 (0.8–11.2) | 372 | 2,988.4 | 10.5 (8.3–13.4) | 178 | 1,996.8 | 8.2 (6.7–10.3) |
| **Region** | | | | | | | | | |
| North | 224 | 1,271.4 | 8.7 (6.8–11.0) | 155 | 737.2 | 9.8 (7.3–13.4) | 69 | 534.2 | 7.5 (5.3–10.3) |
| Center | 308 | 2,184.4 | 9.8 (7.3–12.9) | 211 | 1,399.0 | 11.7 (7.6–17.6) | 97 | 785.3 | 7.6 (5.4–10.5) |
| Mexico City | 129 | 961.1 | 8.3 (5.7–11.8) | 89 | 638.9 | 9.7 (6.5–14.4) | 40 | 322.2 | 6.4 (3.9–10.3) |
| South | 311 | 2,047.9 | 10.2 (8.5–12.4) | 209 | 996.4 | 9.4 (7.4–11.7) | 102 | 1,051.5 | 11.2 (8.3–14.9) |
| **Physical activity[‡]** | | | | | | | | | |
| Inactive | 173 | 1,191.8 | 13.8 (10.5–17.9) | 121 | 576.6 | 11.8 (8.7–15.9) | 52 | 615.2 | 16.3 (10.6–24.2) |
| Moderately active | 81 | 504.3 | 8.9 (6.6–11.9) | 51 | 273.6 | 7.7 (5.1–11.6) | 30 | 230.7 | 10.9 (6.9–16.7) |
| Active | 489 | 3,499.0 | 7.5 (6.1–9.2) | 337 | 2,122.5 | 8.7 (6.4–11.8) | 152 | 1,376.5 | 6.1 (4.8–7.6) |
| **Prior medical diagnosis[§]** | | | | | | | | | |
| High blood pressure | | | | | | | | | |
| No | 514 | 3,357.0 | 5.8(05–6.7) | 338 | 1,841.4 | 6.1(5.2–7.3) | 176 | 1,515.7 | 5.4.(4.3–6.9) |
| Yes | 458 | 3,107.8 | 29.6 (24.2–35.8) | 326 | 1,930.2 | 29.3 (22.3–37.4) | 132 | 1,177.6 | 30.3 (22.6–39.2) |
| Overweight or obesity | | | | | | | | | |
| No | 156 | 1,154.9 | 6.5(4.9–8.6) | 100 | 3,013.4 | 6.6(4.7–9.2) | 56 | 597.3 | 6.4(4.1–9.7) |
| Yes | 756 | 4,912.7 | 10.4 (8.9–12.1) | 534 | 3,013.4 | 11.6 (9.3–14.5) | 222 | 1,899.3 | 8.9 (7.4–10.7) |
| Kidney failure | | | | | | | | | |
| No | 931 | 6,183.7 | 9.1(7.9–10.5) | 635 | 3,577.3 | 9.9 (8.1–12.1) | 296 | 2,606.4 | 8.2(6.8–9.9) |
| Yes | 41 | 281.1 | 38.4 (25.5–53.2) | 29 | 194.2 | 40.6 (24.0–59.7) | 12 | 86.8 | 34.3 (16.9–57.1) |
| Cerebrovascular disease | | | | | | | | | |
| No | 952 | 6,253.2 | 9.2(8.0–10.5) | 652 | 3,642.4 | 10.0(8.2–12.2) | 300 | 2,610.8 | 8.2(6.8–9.9) |
| Yes | 20 | 211.6 | 46.4 (29.3–65.6) | 12 | 129.1 | 51.2 (27.4–74.9) | 8 | 82.4 | 40.2 (17.2–68.5) |
| Acute myocardial infarction or angina | | | | | | | | | |
| No | 902 | 5,948.7 | 8.9(7.7–10.3) | 620 | 3,531.4 | 9.7(8.0–12.1) | 282 | 2,417.3 | 7.8(6.4–9.5) |
| Yes | 70 | 516.1 | 27.7 (19.8–37.3) | 44 | 240.2 | 29.4 (21.2–39.4) | 26 | 276.0 | 26.4 (16.3–39.7) |

[*]Data adjusted for the survey design

[§]Self-report of prior medical diagnosis of the described diseases

[‡] Physical activity level (PA): Inactive <150 minutes a week, moderately active 150–299 minutes a week; and active >300 minutes a week.

**Table 2. Adjusted odds ratio for previously diagnosed diabetes for sociodemographic, anthropometric, and clinical variables.** ENSANUT 2016-México.

| | Total | | | Women | | | Men | | |
|---|---|---|---|---|---|---|---|---|---|
| | **Adjusted OR** | | | **Adjusted OR** | | | **Adjusted OR** | | |
| | **OR** | **(95% CI)** | ***p*** | **OR** | **(95% CI)** | ***p*** | **OR** | **(95% CI)** | ***p*** |
| Sex | | | | | | | | | |
| Man | 1.0 | | | --- | --- | --- | --- | --- | --- |
| Woman | 1.0 | (0.8, 1.4) | *0.916* | --- | --- | --- | --- | --- | --- |
| Age (years) | | | | | | | | | |
| 20–39 | 1.0 | | | 1.0 | | | 1.0 | | |
| 40–59 | 6.3 | (4.1, 9.6) | <0.001 | 4.2 | (2.5, 6.8) | <0.001 | 17.5 | (7.5, 40.6) | <0.001 |
| 60 and more | 13.9 | (8.5, 22.9) | <0.001 | 11.0 | (6.2, 19) | <0.001 | 30.7 | (11.7, 80.5) | <0.001 |
| Socioeconomic tertile | | | | | | | | | |
| Low | 1.0 | | | 1.0 | | | 1.0 | | |
| Medium | 1.0 | (0.7, 1.4) | 0.87 | 0.7 | (0.5, 1) | 0.076 | 1.6 | (0.9, 2.9) | 0.096 |
| High | 0.7 | (0.5, 1.2) | 0.205 | 0.5 | (0.3, 0.9) | 0.017 | 1.4 | (0.7, 2.7) | 0.377 |
| Education level | | | | | | | | | |
| Primary or less | 1.0 | | | 1.0 | | | 1.0 | | |
| Secondary or high school | 1.2 | (0.8, 1.7) | 0.358 | 1.1 | (0.7, 1.6) | 0.801 | 1.3 | (0.7, 2.3) | 0.375 |
| Bachelor's degree | 0.8 | (0.5, 1.2) | 0.287 | 0.5 | (0.3, 1) | 0.036 | 1.5 | (0.7, 3.1) | 0.253 |
| Area of residency | | | | | | | | | |
| Rural | 1.0 | | | 1.0 | | | 1.0 | | |
| Urban | 1.1 | (0.8, 1.6) | 0.476 | 1.4 | (0.9, 2.1) | 0.107 | 0.8 | (0.5, 1.4) | 0.488 |
| Region | | | | | | | | | |
| North | 1.0 | | | 1.0 | | | 1.0 | | |
| Center | 1.1 | (0.8, 1.7) | 0.539 | 1.2 | (0.7, 2) | 0.56 | 1 | (0.5, 1.9) | 0.968 |
| Mexico City | 0.8 | (0.5, 1.3) | 0.356 | 0.8 | (0.4, 1.3) | 0.317 | 0.9 | (0.4, 1.8) | 0.707 |
| South | 1 | (0.7, 1.4) | 0.819 | 0.7 | (0.5, 1.2) | 0.176 | 1.4 | (0.8, 2.7) | 0.275 |
| Body Mass Index | | | | | | | | | |
| Normal | 1.0 | | | 1.0 | | | 1.0 | | |
| Overweight | 1.4 | (0.9, 2) | 0.136 | 1.1 | (0.6, 1.7) | 0.824 | 1.7 | (1.1, 3.0) | 0.051 |
| Obesity | 1.2 | (0.8, 1.7) | 0.351 | 1.1 | (0.7, 1.8) | 0.715 | 1.2 | (0.7, 2) | 0.593 |
| Prior medical diagnosis§ | | | | | | | | | |
| No | 1.0 | | | 1.0 | | | 1.0 | | |
| High blood pressure | 3.3 | (2.4, 4.4) | <0.001 | 2.9 | (2.0, 4.1) | <0.001 | 4.2 | (2.6, 7) | <0.001 |
| Cerebrovascular disease | 2.8 | (1.1, 6.9) | 0.027 | 3.8 | (1.4, 10.6) | 0.011 | 1.3 | (0.2, 9.8) | 0.816 |
| Acute myocardial infarction | 1.4 | (0.8, 2.7) | 0.276 | 1.7 | (0.8, 3.8) | 0.182 | 1.4 | (0.6, 3.7) | 0.45 |
| Kidney failure | 3.5 | (1.8, 7.1) | <0.001 | 5.1 | (2.5, 10.7) | <0.001 | 1.8 | (0.4, 7.2) | 0.436 |
| Physical activity‡ | | | | | | | | | |
| Inactive | 1.0 | | | 1.0 | | | 1.0 | | |
| Moderately active | 0.9 | (0.6, 1.4) | 0.616 | 0.7 | (0.4, 1.3) | 0.288 | 1.2 | (0.6, 2.7) | 0.587 |
| Active | 0.8 | (0.5, 1.1) | 0.198 | 0.8 | (0.5, 1.2) | 0.212 | 0.7 | (0.4, 1.3) | 0.258 |
| Screen time (minutes) | | | | | | | | | |
| ≤840 | 1.0 | | | 1.0 | | | 1.0 | | |
| 840–1680 | 1.0 | (0.7, 1.5) | 0.883 | 1.3 | (0.9, 1.9) | 0.149 | 0.7 | (0.4, 1.1) | 0.124 |
| 1680 or more | 1.0 | (0.6, 1.6) | 0.908 | 1.3 | (0.8, 2.4) | 0.324 | 0.5 | (0.2, 1.2) | 0.105 |
| Dietary diversity | | | | | | | | | |
| First quartile | 1.0 | | | 1.0 | | | 1.0 | | |
| Second quartile | 0.7 | (0.5, 1.1) | 0.104 | 0.6 | (0.4, 0.9) | 0.026 | 1.2 | (0.6, 2.3) | 0.578 |
| Third quartile | 0.7 | (0.4, 1.1) | 0.068 | 0.6 | (0.3, 0.9) | 0.022 | 1.2 | (0.6, 2.3) | 0.687 |
| Fourth quartile | 0.5 | (0.3, 0.7) | <0.001 | 0.4 | (0.2, 0.7) | 0.001 | 0.7 | (0.4, 1.3) | 0.228 |

(*Continued*)

**Table 2.** (Continued)

| | Total | | | Women | | | Men | | |
|---|---|---|---|---|---|---|---|---|---|
| | **Adjusted OR** | | | **Adjusted OR** | | | **Adjusted OR** | | |
| | OR | (95% CI) | *p* | OR | (95% CI) | *p* | OR | (95% CI) | *p* |
| Has smoked more than 100 cigarettes | | | | | | | | | |
| Never smoked | 1.0 | | | 1.0 | | | 1.0 | | |
| Has never smoked more than 100 cigarettes | 0.8 | (0.6, 1.2) | 0.36 | 0.8 | (0.5, 1.3) | 0.342 | 0.8 | (0.5, 1.4) | 0.432 |
| Has smoked more than 100 but not smoking anymore | 1.1 | (0.7, 1.6) | 0.74 | 0.9 | (0.4, 2) | 0.781 | 1.1 | (0.6, 1.8) | 0.87 |
| Has smoked more than 100 and is still smoking | 0.7 | (0.4, 1.1) | 0.107 | 0.7 | (0.3, 1.3) | 0.197 | 0.7 | (0.4, 1.4) | 0.336 |
| Sleep quality | | | | | | | | | |
| Good to fair | 1.0 | | | 1.0 | | | 1.0 | | |
| Bad or very bad | 1.0 | (0.7, 1.5) | 0.913 | 1.1 | (0.7, 1.8) | 0.692 | 1.0 | (0.5, 2) | 0.964 |

*Data adjusted for the survey design

Adjusted OR for sociodemographic variable (sex, age, socioeconomic tertile, education level, area of residency and region); anthropometric (body mass index); prior medical diagnosis (high blood pressure, cerebrovascular disease, acute myocardial infarction or angina and kidney failure); and lifestyles (physical activity, screen time, dietary diversity, smoking and sleep quality).

§ Self-report of prior medical diagnosis of the described diseases

‡ Physical activity(PA) level: Inactive <150 minutes a week, moderately active 150–299 minutes a week; and active >300 minutes a week.

kidney failure was associated with a higher OR for having diabetes (p<0.05). When we compare the diversity of consumption of food groups or DD, we observe that in the total population and women with the highest quintile of DD (fourth quintile) the OR of having diabetes was lower (total population 0.5 CI 95% 0.3–0.7; women 0.4 CI 95% 0.2–0.7) than in the first quintile (OR = 1.0). In an adjusted model we tested interactions between each of the included variables and sex, observing significant interaction (p <0.055) only with age, socioeconomic tertile and education level.

A total of 30.5% of adults with diabetes did not report any control strategies, 44.9% measured their venous blood glucose, and 15.2% used the HbA1C as an indicator of glycemic control (Table 3). Only 46.4% of them reported preventive measures. When comparing by sex groups, frequency of when the glycemic control assessment was performed during the last year, measures to prevent complications, and lifestyle interventions, there were no differences when categorizing by sex, except when comparing how frequently the dental evaluation was performed: women self-reported 2.6 times more this practice than men did (11.3% vs 4.3%).

## Discussion

In our analysis, we found that in Mexican adults the prevalence of previously diagnosed diabetes was 9.4% and only 44.9% used some glycemic control strategy. The prevalence of diabetes in Mexico is higher than in countries like Holland [19] (5.4%) and the average prevalence in the world (8.5%) [20]; probably because the prevalence of overweight in Mexico is at the top of the worldwide rankings and it is the main precipitant factor. [21] The prevalence of previous diagnosis of diabetes in Mexico increased from 7.0 to 9.2% between 2006 and 2012, [8] however, in the following four years (2016) the increase was only 0.2%. [11] This reflects that fewer people with diabetes are unaware of having this disease and the timely diagnosis has improved in recent years.

Diabetes occurs mainly in persons in their fourth decade, [22] and in our results, we found that the prevalence was higher in adults aged 40 years and older. This trend is similar to the

**Table 3. Strategies for assessing glycemic control during the last year and measures to prevent complications associated with diabetes, categorizing by sex.** ENSA-NUT MC 2016*.

| | Total | | | Women | | | Men | | |
|---|---|---|---|---|---|---|---|---|---|
| | n | % | (95% CI) | n | % | (95% CI) | n | % | (95% CI) |
| **Glycemic control assessment** | | | | | | | | | |
| Urine reactive strips | 60 | 4.6 | (3.1–7.0) | 36 | 3.1 | (1.9–5.2) | 24 | 6.7 | (3.7–12.1) |
| Blood reactive strips | 242 | 22.3 | (17.7–27.7) | 156 | 18.1 | (13.6–25.6) | 86 | 28.2 | (20.7–27.2) |
| Urinalysis | 308 | 29.6 | (24.9–34.7) | 227 | 34.2 | (27.7–41.2) | 81 | 23.1 | (17.3–30.2) |
| Venous blood sampling | 462 | 44.9 | (28.2–51.7) | 333 | 48.4 | (39.9–57.2) | 129 | 39.8 | (30.6–49.1) |
| HbA1c testing | 153 | 15.2 | (11.7–19.5) | 122 | 17.5 | (13.1–22.9) | 31 | 12.1 | (7.1–19.1) |
| Protein in urine | 44 | 4.6 | (3.2–6.8) | 27 | 3.9 | (2.2–6.8) | 17 | 5.7 | (3.3–9.7) |
| Self-monitoring/self-management | 19 | 1.7 | (0.9–3.1) | 12 | 1.3 | (0.7–2.7) | 7 | 2.3 | (0.9–5.5) |
| No testing | 716 | 30.5 | (24.7–29.3) | 499 | 30.5 | (21.8–22.1) | 217 | 30.6 | (22.1–40.7) |
| **Preventive measures** | | | | | | | | | |
| Eye exam | 130 | 13.1 | (10.4–16.4) | 97 | 15.3 | (11.7–19.6) | 33 | 10.1 | (6.5–15.2) |
| Cholesterol and triglyceride measurement | 155 | 15.2 | (12.3–18.7) | 112 | 15.4 | (11.3–20.7) | 43 | 14.9 | (10.6–20.6) |
| Blood pressure measurement | 43 | 6.1 | (3.2–10.9) | 26 | 4.3 | (2.4–7.5) | 17 | 8.5 | (3.2–20.5) |
| Kidney exam/microalbuminuria | 140 | 14.2 | (11.5–17.5) | 103 | 15.1 | (11.6–19.4) | 37 | 13.1 | (8.9–18.5) |
| Electrocardiogram | 42 | 4.4 | (2.8–6.9) | 26 | 3.5 | (1.8–7.0) | 16 | 5.6 | (3.2–9.8) |
| Taking a daily aspirin | 61 | 5.1 | (3.53–7.41) | 46 | 7.2 | (4.6–11.1) | 15 | 2.3 | (1.1–4.7) |
| Influenza-pneumococcal yearly immunizations | 165 | 15.3 | (12.3–18.9) | 115 | 16.3 | (12.3–21.6) | 50 | 13.9 | (10.1–19.1) |
| Dental exam | 93 | 8.4 | (6.2–11.2) | 75 | 11.3 | (7.8–16.1) | 18 | 4.3 | (2.7–6.9) |
| No preventive measure | 503 | 53.6 | (46.6–60.4) | 345 | 52.1 | (43.2–60.7) | 158 | 55.8 | (46.5–64.8) |
| **Lifestyle interventions** | | | | | | | | | |
| Educational diabetes program | 111 | 9.3 | (6.9–12.2) | 86 | 10.6 | (7.5–14.7) | 25 | 7.3 | (4.5–11.8) |
| Quit smoking | 19 | 2.4 | (1.4–4.2) | 8 | 1.8 | (0.8–3.9) | 11 | 3.2 | (1.5–6.8) |
| Avoid shoes that injure feet | 80 | 8.4 | (5.6–12.5) | 50 | 7.7 | (4.9–11.9) | 30 | 9.4 | (5.49–15.8) |
| Physical activity | 570 | 77.1 | (69.7–83.1) | 388 | 80.6 | (73.1–86.4) | 182 | 72.3 | (59.5–82.3) |

*Data adjusted for the survey design

one in the U.S., because diabetes can be the result of a culmination of health problems that accumulate throughout life. [23]

It has been described that the excess of body fat is a factor tightly related to the development of insulin resistance and later to diabetes. [20] We found that adults with obesity had a higher probability ratio for being diagnosed with diabetes (OR 1.8, 95% CI 1.3–2.8) than adults with normal BMI do.

Diabetes can be a reflection of the behavioral, hereditary, and social context risk factors. Those who belong to the lowest SES tertile have a higher risk of developing diabetes because they have less access to health services, to prompt diagnosis, and to a healthy lifestyle. [24] We found that adults from a low SES had a higher probability ratio for being diagnosed with diabetes (OR 1.0) than those from a higher SES (OR 0.6, 95% CI 0.4–0.9).

One of the aims of this study was to describe which glycemic control strategies were used more frequently in Mexico. Venous blood glucose measurement and glycosylated hemoglobin (HbA1c) quantification are the glycemic control strategies recommended by the American Diabetes Association (ADA). [25] Our findings showed that 30.5% of the Mexicans with diabetes did not have any control strategies, that 44.9% measured their venous blood glucose, and only 15.2% used the HbA1c as an indicator for glycemic control during the last 12 months. Although there are no statistics of these indicators in other countries, in the National Health

and Nutrition Examination Survey III, only 39% of the participants reported using glycemic screening. [26]

The ADA has established strategies for comorbidities prevention. These include measuring blood pressure, cholesterol and triglycerides in blood, protein in urine; eye and teeth evaluations, as well as applying immunizations. [25] In the ENSANUT-2016 we observed that 53.6% of the population did not take any preventive measures. Even though 49.2% of Mexican adults had hypertension. [27] Of the adults analyzed in this study, only 6.1% verified their blood pressure; therefore, it is possible that there is a high percentage of adults with hypertension who are unaware of having this disease. This would increase the risk of developing associated complications if this situation is not reversed in the short term.

As for lifestyle interventions, the ADA recommends performing PA, not smoking, and improving diet, among other. [28] In the ENSANUT-2016, we found that PA was the most practiced intervention (77.1%) to control glycemia and to prevent the development of comorbidities. This figure is similar to the one reported in persons without diabetes but could be overestimated due to the questionnaire used. [29]

Dietary management is important to prevent diabetes [30] and dietary diversity is inversely associated with the risk of developing diabetes. [31] In our analysis we found that in women and total population having a greater DD was associated with a lower probability ratio of being diagnosed with diabetes. In men we do not find that DD is associated with a lower risk of diabetes possibly because in some subpopulations such as Hispanics [32] the results are still inconsistent and it is necessary to use a methodology that measures DD more accurately. These findings should motivate the generation of new studies that analyze this association longitudinally to confirm the direction and magnitude of causality. If this association is confirmed, it would be advisable to design communication programs to promote DD as another strategy to prevent diabetes.

According to ADA's standards, all people with diabetes should participate in a self-management educational program. In this study we show that only 9% of the adults with diabetes received an education to facilitate the knowledge and to improve self-management skills. This explains in part why half of the adults do not take a preventive measure.

Some of the limitations of our analysis are that due to the study design we could not establish causality with risk factors and we could not know the percentage of adults who had diabetes but have not been diagnosed yet. We recognize that our results may be influenced by the possible measurement bias that represents the use of a self-report and by the social desirability bias in answering the questionnaires, however, this measurement tool has a high sensitivity and specificity as an indicator of prior medical diagnosis of diabetes. [33–34]

Not having glucose measurement as a complementary diagnostic method may underestimate the true prevalence of diabetes by up to 50%. [35] For example, in Mexican immigrants participating in the National Health and Nutrition Examination Survey when self-report is used as a diagnostic tool, only half of adults with diabetes (11.3%) are detected compared to using glucose measurement as a complementary method (22.6%). [36]

One of the strengths of the study is that the results are representative of the Mexican adult population and are the most recent data on the prevalence of diabetes. This information will help the decision makers in health policies to know the magnitude of this disease, main associated risk factors and diabetes control practices.

The high prevalence of diabetes found in our study should motivate in the short term estimate the total prevalence of diabetes including glucose measurement as a diagnostic method. We also believe it is necessary to evaluate the suitability of current programs for the diagnosis, prevention and control of diabetes such as PrevenIMSS and PrevenISSSTE to reduce its prevalence and improve glycemic control.

## Conclusion

The conclusion of this study is that the prevalence of previously diagnosed diabetes among Mexican adults who participated in the ENSANUT-2016 was high. Being older or obese are risk factors that increase the probability ratio for an adult being diagnosed with diabetes. Finally, approximately half of Mexican adults with diabetes implement strategies to assess glycemic control and to prevent complications.

## Author Contributions

**Conceptualization:** Eric Monterrubio-Flores.

**Data curation:** Andrys Valdez.

**Formal analysis:** Ismael Campos-Nonato, Eric Monterrubio-Flores.

**Investigation:** Ismael Campos-Nonato, Alejandra Flores-Coria.

**Methodology:** Ismael Campos-Nonato, María Ramírez-Villalobos, Eric Monterrubio-Flores.

**Software:** Eric Monterrubio-Flores.

**Supervision:** Ismael Campos-Nonato.

**Validation:** Ismael Campos-Nonato, María Ramírez-Villalobos, Alejandra Flores-Coria, Andrys Valdez.

**Writing – original draft:** Ismael Campos-Nonato, María Ramírez-Villalobos.

**Writing – review & editing:** Ismael Campos-Nonato, María Ramírez-Villalobos, Alejandra Flores-Coria, Andrys Valdez, Eric Monterrubio-Flores.

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
