## [Decision Letter · Decision Letter 0]

10 Dec 2019

PONE-D-19-29166

Prevalence of previously diagnosed diabetes and glycemic control strategies in Mexican adults: ENSANUT-2016

PLOS ONE

Dear M.Sc Ramirez-Villalobos,

Thank you for submitting your manuscript to PLOS ONE. After careful consideration, we feel that it has merit but does not fully meet PLOS ONE’s publication criteria as it currently stands. Therefore, we invite you to submit a revised version of the manuscript that addresses the points raised during the review process.

We would appreciate receiving your revised manuscript by Jan 24 2020 11:59PM. To enhance the reproducibility of your results, we recommend that if applicable you deposit your laboratory protocols in protocols.io, where a protocol can be assigned its own identifier (DOI) such that it can be cited independently in the future. For instructions see: http://journals.plos.org/plosone/s/submission-guidelines#loc-laboratory-protocols

We look forward to receiving your revised manuscript.

Kind regards,

Naeti Suksomboon

Academic Editor

PLOS ONE

Journal Requirements:

1.

3. Thank you for stating the following financial disclosure: "No"

Please provide an amended Funding Statement that declares *all* the funding or sources of support received during this specific study (whether external or internal to your organization) as detailed online in our guide for authors at http://journals.plos.org/plosone/s/submit-now.  Please state what role the funders took in the study.  If any authors received a salary from any of your funders, please state which authors and which funder. If the funders had no role, please state: "The funders had no role in study design, data collection and analysis, decision to publish, or preparation of the manuscript."

Reviewers' comments:

Reviewer's Responses to Questions

**Comments to the Author**

1. Is the manuscript technically sound, and do the data support the conclusions?

Reviewer #1: Partly

2. Has the statistical analysis been performed appropriately and rigorously? 

Reviewer #1: I Don't Know

3. Have the authors made all data underlying the findings in their manuscript fully available?

Reviewer #1: Yes

4. Is the manuscript presented in an intelligible fashion and written in standard English?

Reviewer #1: Yes

5. Review Comments to the Author

Reviewer #1: This study provides estimates of diabetes among Mexican adults using data from the most recent national health survey in Mexico. The topic of the study is of high importance, given how prominent diabetes and associated risk factors, like overweight and obesity, are in Mexico and the significant health, societal and economic impacts these conditions have at present.

The paper can be improved by clarifying certain methodological aspects and assertions, better organizing of the results,by situating the findings in the context of any national diabetes prevention and control strategies, and adding an implications/recommendations for research and/or practice paragraph to the discussion.

Abstract

• There is a typo on line 26, pertaining to the year of the ENSANUT survey.

• Unclear whether odds ratios are crude or adjusted for other factors.

• On lines 34-36, it is unclear whether the percentages reported refer to all ENSANUT participants or only those diagnosed with diabetes. Also, if 44.9% measured their venous blood glucose, how is it possible that 69.5% were estimated as not engaging in any control strategies? The two percentages would exceed 100%.

• The discussion should be based on reported results, not report new results. The authors mention activity and education, but those results are not reported in the results section.

• The discussion also mentions that half of partipants used control strategies but that percentage is not reported in the results section either.

Introduction

• It should be made clear whether the paper focuses on Diabetes Type 2 or any type of diabetes.

• Why is the ENSANUT 2012 reported as the last national health survey? Shouldn’t it be ENSANUT 2016?

Methods

• Line 71: I don’t think the word “considering” is appropriate here. I think the authors probably mean that the survey “achieved” a 91.7% response rate. Please, double check.

• Line 86: Did these questions inquire about lifetime diagnosis or within a certain time window?

• Lines 106-109: The decription of the DD score categorizatio into recommended and non-recommended foods is confusing. These dietary measures need to be explained more clearly. Perhaps some examples would help.

• Lines 110-112: These measures of sleep and glycemic control strategies require more elaboration to increase the rigor of the study. The authors also need to specify if questions about glycemic control strategies were asked of all participants or only those who reported a diabetes diagnosis. If only those with diabetes were administered these questions, the n sizes should be added to the first row of Table 3. If this information was obtained from everyone, it would make sense to stratify the results about glycemic control by previous diagnosis of diabetes, as they can expect to vary significantly by disease status.

Statistical Analyses

• Please, unpack the variables used for adjustment, to increase replicability of the study. The list of variables adjusted for should also be reported as a footnote under Table 2.

Results/Tables

• It appears, per the 95% CI, that women with primary or less education were also more likely to be diagnosed with diabetes than their male counterparts, but this result is not reported.

• The result reported for gender differences among those with overweight or obesity (Lines 128-129) seems at odds with the overlapping 95% CIs shown on Table 1. Is that result after adjustment for other factors? If so, this should be stated.

• Table 1 should show the prevalence rates among those with and without the conditions listed under “Prior Medical Diagnosis”. That is, prevalence for those with and without a diagnosis of high blood pressure; with and without overweight or obesity; etc.

• The reporting of results shown on Table 2 (Lines 145-151) can be significantly improved. I recommend reporting the results separately for women and men and being more systematic listing all of the factors that were statistically associated with a diabetes diagnosis after adjustiment for other confounders, instead of picking/choosing just a few.

• I would like to see the results for all (women and men combined) on Table 2, just like Tables 1 and 3. If it’s difficult to fit all of the information on one table, the authors could consider presenting only adjusted results and provide the unadjusted ORs as supplemental material?

• On Table 3, it is unclear what “no testing” means. As stated in one of my comments for the abstract, if 69.5% did not testing, how is it possible that 44.9% did venous blood sampling? This should be mutually exclusive categories, yet they add up to more than 100%.

Discussion

• Please, comment on how the estimated prevalence compares to estimates based on earlier ENSANUT surveys, to give a sense of any potential trends.

• Lines 207-208: Please, double check and clarify why these percents exceed 100%.

• Lines 211-212: Specify this statistic is for the U.S.

• Lines 219-220: Clarify what “sub-diagnosis” means here and revise this sentence to make its meaning clearer. Is this about hypertension or about diabetes?

• Lines 224-225: This sentence is unclear. Is this still based on ENSANUT 2016? Also, the issue of social desirability bias should be addressed separately, as part of the limitations, as it can apply to all of the self-report based data used for this analysis.

• Line 231-233: The authors could cite other studies that have compared measured versus self-reported diabetes for Mexican adults, even if outside Mexico, as further evidence of likely undestimatio of true prevalence of diabetes. See Barcellos et al. Health Affairs 2012;31(12).

• A paragraph with implications and/or recommendations for future research and practice would strengthen the discussion, as would adding a little bit of context regarding any national strategies to improve diabetes prevention and control.

6. PLOS authors have the option to publish the peer review history of their article (what does this mean?). If published, this will include your full peer review and any attached files.

Reviewer #1: No

---

## [Author Response · Author response to Decision Letter 0]

29 Jan 2020

PONE-D-19-29166

Prevalence of previously diagnosed diabetes and glycemic control strategies in Mexican adults: ENSANUT-2016

Comments to the author.

1. Is the manuscript technically sound, and do the data support the conclusions?

Reviewer #1: Partly

2. Has the statistical analysis been performed appropriately and rigorously? 

Reviewer #1: I Don't Know

3. Have the authors made all data underlying the findings in their manuscript fully available?

Reviewer #1: Yes

 4. Is the manuscript presented in an intelligible fashion and written in standard English?

Reviewer #1: Yes

Reviewer´s comment 1. Abstract: There is a typo on line 26, pertaining to the year of the ENSANUT survey.

Authors response: We appreciate the observation. We have corrected the name of the survey and now "ENSANUT-2016" appears.

Reviewer´s comment 2. Unclear whether odds ratios are crude or adjusted for other factors.

Authors’ response: We clarify that in Table 2, in the last row it is described that OR are “adjusted for sociodemographic, anthropometric and clinical variables from the table”.

In the abstract, we now specify that the OR are adjusted: “The adjusted OR for having diabetes was higher in adults aged ≥60 years (OR = 11.0 in women and OR = 30.7 in men) than in adults aged 20-39 years (OR=1.0). The adjusted OR for having diabetes was higher in overweight men (OR=1.7) than in men with normal BMI (OR=1.0).”

Reviewer´s comment 3. On lines 34-36, it is unclear whether the percentages reported refer to all ENSANUT participants or only those diagnosed with diabetes. Also, if 44.9% measured their venous blood glucose, how is it possible that 69.5% were estimated as not engaging in any control strategies? The two percentages would exceed 100%.

Authors’ response: We appreciate the observation. We have corrected the data and now the following appears: “A total of 30.5% of the participants did not report any control strategies, 44.9% measured their venous blood glucose...,”.

Reviewer´s comment 4. The discussion should be based on reported results, not report new results. The authors mention activity and education, but those results are not reported in the results section.

Authors’ response: The authors have attended this comment. Now the risk factors described in the results section are the same as those included in the conclusions section (age and overweight).

Reviewer´s comment 5. The discussion also mentions that half of participants used control strategies but that percentage is not reported in the results section either.

Authors’ response: In response to this observation in results section we add: “A total of 30.5% of adults with diabetes did not report any control strategies, 44.9% measured their venous blood glucose, and 15.2% used the HbA1C as an indicator of glycemic control”

Reviewer´s comment 6. Introduction. It should be made clear whether the paper focuses on Diabetes Type 2 or any type of diabetes.

Authors’ response: Now we specify that the manuscript is focused on type 2 diabetes

Reviewer´s comment 7. Introduction. Why is the ENSANUT 2012 reported as the last national health survey? Shouldn’t it be ENSANUT 2016?.

Authors’ response: We appreciate the observation and to avoid confusion we have now written it as follows: "according to the national health survey 2012 ..."

Reviewer´s comment 8. Methods. Line 71: I don’t think the word “considering” is appropriate here. I think the authors probably mean that the survey “achieved” a 91.7% response rate. Please, double check.

Authors’ response: We appreciate the suggestion and now we include in the text "achieved”

Reviewer´s comment 9. Methods. Line 86: Did these questions inquire about lifetime diagnosis or within a certain time window?

Authors’ response: Now we specify in methods that the diagnosis of these pathologies refers to “throughout life”.

Reviewer´s comment 10. Methods. Lines 106-109: The description of the DD score categorization into recommended and non-recommended foods is confusing. These dietary measures need to be explained more clearly. Perhaps some examples would help.

Authors’ response: Now we specify: “... recommended (for example; fruits, vegetables, legumes, tubers, cereals with fiber, dairy, sugar-free drinks) and non-recommended foods (cereals with sugar, sweets and desserts, sugary drinks, alcoholic beverages, dairy drinks with sugar and processed meats).”

Reviewer´s comment 11. Methods. Lines 110-112: The authors also need to specify if questions about glycemic control strategies were asked of all participants or only those who reported a diabetes diagnosis. If only those with diabetes were administered these questions, the n sizes should be added to the first row of Table 3. If this information was obtained from everyone, it would make sense to stratify the results about glycemic control by previous diagnosis of diabetes, as they can expect to vary significantly by disease status.

Authors’ response: We have added the description that the questions about glycemic control “were self-reported only by participants previously diagnosed with type 2 diabetes”. On the other hand, Table 3 includes the “n” of each category.

Reviewer´s comment 12. Statistical Analyses. Please, unpack the variables used for adjustment, to increase replicability of the study. The list of variables adjusted for should also be reported as a footnote under Table 2.

Authors’ response: Now in Table 2 the variables that were used to adjust are shown as footnotes: “Adjusted OR for sociodemographic variable (sex, age, socioeconomic tertile, education level, area of residency and region); anthropometric (body mass index); prior medical diagnosis (high blood pressure, cerebrovascular disease, acute myocardial infarction or angina and kidney failure); and lifestyles (physical activity, screen time, dietary diversity, smoking and sleep quality).

Reviewer´s comment 13. Results/Tables. It appears, per the 95% CI, that women with primary or less education were also more likely to be diagnosed with diabetes than their male counterparts, but this result is not reported.

Authors’ response: Now we describe in the manuscript the following: " Women with primary or less education were also more likely to be diagnosed with diabetes (18.8%; CI95% 14.9-23.4) than their male counterparts (12.5%; CI95% 10.4-14.8)”.

Reviewer´s comment 14. Results/Tables. The result reported for gender differences among those with overweight or obesity (Lines 128-129) seems at odds with the overlapping 95% CIs shown on Table 1. Is that result after adjustment for other factors? If so, this should be stated.

Authors’ response: We appreciate the observation. Because 95% CIs overlap, we decided to exclude that description: “When identifying adults with overweight or obesity, the prevalence of diabetes was 30% higher in women than in men”.

Reviewer´s comment 15. Results/Tables. Table 1 should show the prevalence rates among those with and without the conditions listed under “Prior Medical Diagnosis”. That is, prevalence for those with and without a diagnosis of high blood pressure; with and without overweight or obesity; etc.

Authors’ response: Now the Table 1 shows the prevalence with or without the disease.

Reviewer´s comment 16. Results/Tables. The reporting of results shown on Table 2 (Lines 145-151) can be significantly improved. I recommend reporting the results separately for women and men and being more systematic listing all of the factors that were statistically associated with a diabetes diagnosis after adjustment for other confounders, instead of picking/choosing just a few.

Authors’ response: The authors appreciate the suggestion, however, we believe that the description included shows the most relevant findings. To provide more detail we have added the result of interactions: “In an adjusted model we tested interactions between each of the included variables and sex, observing significant interaction (p <0.055) only with age, socioeconomic tertile and education level."

Reviewer´s comment 17. Results/Tables. I would like to see the results for all (women and men combined) on Table 2, just like Tables 1 and 3. If it’s difficult to fit all of the information on one table, the authors could consider presenting only adjusted results and provide the unadjusted ORs as supplemental material?

Authors’ response: We attended this comment and now we include the combined results in Tables 1, 2 and 3.

Reviewer´s comment 18. Results/Tables. On Table 3, it is unclear what “no testing” means. As stated in one of my comments for the abstract, if 69.5% did not testing, how is it possible that 44.9% did venous blood sampling? This should be mutually exclusive categories, yet they add up to more than 100%.

Authors’ response: We appreciate the valuable comment. We review each value in the table and now we include the correct data: “... A total of 30.5% of adults with diabetes did not report any control strategies, 44.9% measured their venous blood glucose...”

Reviewer´s comment 19. Discussion. Please, comment on how the estimated prevalence compares to estimates based on earlier ENSANUT surveys, to give a sense of any potential trends.

Authors’ response: We have added the following information: “The prevalence of previous diagnosis of diabetes in Mexico increased from 7.0 to 9.2% between 2006 and 2012,[8] however, in the following six years (2016) the increase was only 0.2%.[11] This reflects that fewer people with diabetes are unaware of having this disease and the timely diagnosis has improved in recent years”.

Reviewer´s comment 20. Discussion. Lines 207-208: Please, double check and clarify why these percents exceed 100%.

Authors’ response: We appreciate the valuable comment. We review each value in the table and now we include the correct data. The percentage described before was 69.5%, but the correct was the complement (30.5%).

Reviewer´s comment 21. Discussion. Lines 219-220: Clarify what “sub-diagnosis” means here and revise this sentence to make its meaning clearer. Is this about hypertension or diabetes?

Authors’ response: To avoid confusion we have changed the description as follows: “...Even though 49.2% of Mexican adults had hypertension. [27] Of the adults analyzed in this study, only 6.1% verified their blood pressure; therefore, it is possible that there is a high percentage of adults with hypertension and are unaware of having this disease. This would increase the risk of developing associated complications if this situation is not reversed in the short term.

Reviewer´s comment 22. Discussion. Lines 224-225: This sentence is unclear. Is this still based on ENSANUT 2016? Also, the issue of social desirability bias should be addressed separately, as part of the limitations, as it can apply to all of the self-report based data used for this analysis.

Authors’ response: To avoid confusion: 

1) We now specify that the result is based on ENSANUT 2016.

2) The reference to social desirability bias was moved to the limitations section. Now we describe this as follows: “We recognize that our results may be influenced by the possible measurement bias that represents the use of a self-report and by the social desirability bias in answering the questionnaires”.

Reviewer´s comment 23. Discussion. Line 231-233: The authors could cite other studies that have compared measured versus self-reported diabetes for Mexican adults, even if outside Mexico, as further evidence of likely undestimatio of true prevalence of diabetes. See Barcellos et al. Health Affairs 2012;31(12). 

Authors’ response: We appreciate the suggestion of the reference and the comment. 

We have now added the following paragraph to the discussion: "... in Mexican immigrants participating in the National Health and Nutrition Examination Survey when self-report is used as a diagnostic tool, only half of adults with diabetes (11.3%) are detected compared to using glucose measurement as a complementary method (22.6%).[33]

Reviewer´s comment 24. Discussion. A paragraph with implications and/or recommendations for future research and practice would strengthen the discussion, as would adding a little bit of context regarding any national strategies to improve diabetes prevention and control. 

Authors’ response: To respond we have added the following paragraph in the discussion: “The high prevalence of diabetes found in our study should motivate in the short term estimate the total prevalence of diabetes including glucose measurement as a diagnostic method. We also believe it is necessary to evaluate the suitability of current programs for the diagnosis, prevention and control of diabetes such as PrevenIMSS and PrevenISSSTE to reduce its prevalence and improve glycemic control”.

---

## [Decision Letter · Decision Letter 1]

24 Feb 2020

PONE-D-19-29166R1

Prevalence of previously diagnosed diabetes and glycemic control strategies in Mexican adults: ENSANUT-2016

PLOS ONE

Dear M.Sc Ramirez-Villalobos,

Thank you for submitting your manuscript to PLOS ONE. After careful consideration, we feel that it has merit but does not fully meet PLOS ONE’s publication criteria as it currently stands. Therefore, we invite you to submit a revised version of the manuscript that addresses the points raised during the review process.

Kindly address reviewer comment.

We would appreciate receiving your revised manuscript by Apr 09 2020 11:59PM. To enhance the reproducibility of your results, we recommend that if applicable you deposit your laboratory protocols in protocols.io, where a protocol can be assigned its own identifier (DOI) such that it can be cited independently in the future. For instructions see: http://journals.plos.org/plosone/s/submission-guidelines#loc-laboratory-protocols

We look forward to receiving your revised manuscript.

Kind regards,

Naeti Suksomboon

Academic Editor

PLOS ONE

Reviewers' comments:

Reviewer's Responses to Questions

**Comments to the Author**

1. If the authors have adequately addressed your comments raised in a previous round of review and you feel that this manuscript is now acceptable for publication, you may indicate that here to bypass the “Comments to the Author” section, enter your conflict of interest statement in the “Confidential to Editor” section, and submit your "Accept" recommendation.

Reviewer #1: All comments have been addressed

2. Is the manuscript technically sound, and do the data support the conclusions?

Reviewer #1: Yes

3. Has the statistical analysis been performed appropriately and rigorously? 

Reviewer #1: Yes

4. Have the authors made all data underlying the findings in their manuscript fully available?

Reviewer #1: Yes

5. Is the manuscript presented in an intelligible fashion and written in standard English?

Reviewer #1: Yes

6. Review Comments to the Author

Reviewer #1: The authors have addressed most of my previous comments adequately and I think this revised version is a fine contribution to the literature. I only have a remaining concern regarding the dietary measure and a few, very minor additional comments that I would like to see addressed:

• The description of the dietary measure is still unclear. I understand how the initial DD score was computed (number of food groups x number of days they were consumed) and the use of quartiles to create four groups from lowest to highest dietary diversity, but I am confused about the classification of food groups into desirable and not desirable. How was the classification of the food groups into these categories factored in when computing the DD score and quartiles?

• I would like to see the findings regarding dietary diversity as a protective factor against a diabetes diagnoses reported in the results section and commented on the discussion. It seems to me this is an important result with implications for public health campaigns promoting dietary improvements. It is also interesting that the pattern of results for this factor is different for men and women, with a significant association for women, but not for men.

• On Table 1, the estimates for individuals without kidney failure are missing.

• On line 195, the “six” years appears incorrect. It should be 10 (if the reference point is 2006) or 4 (if the reference point is 2012).

• On line 227, there seems to be a typo. I think the authors mean “…with hypertension who are unaware”.

• The sentence on lines 259-261 needs to be revised to make sense.

7. PLOS authors have the option to publish the peer review history of their article (what does this mean?). If published, this will include your full peer review and any attached files.

Reviewer #1: No

---

## [Author Response · Author response to Decision Letter 1]

4 Mar 2020

RESPONSE TO REVIEWER'S COMMENTS

PONE-D-19-29166R1

Prevalence of previously diagnosed diabetes and glycemic control strategies in Mexican adults: ENSANUT-2016

Comments to the author.

1. If the authors have adequately addressed your comments raised in a previous round of review and you feel that this manuscript is now acceptable for publication, you may indicate that here to bypass the “Comments to the Author” section, enter your conflict of interest statement in the “Confidential to Editor” section, and submit your "Accept" recommendation.

Reviewer #1: All comments have been addressed

2. Has the statistical analysis been performed appropriately and rigorously? 

Reviewer #1: Yes

3. Have the authors made all data underlying the findings in their manuscript fully available?

Reviewer #1: Yes

 4. Is the manuscript presented in an intelligible fashion and written in standard English?

Reviewer #1: Yes

Comments to the Author, 1. 

The description of the dietary measure is still unclear. I understand how the initial DD score was computed (number of food groups x number of days they were consumed) and the use of quartiles to create four groups from lowest to highest dietary diversity, but I am confused about the classification of food groups into desirable and not desirable. How was the classification of the food groups into these categories factored in when computing the DD score and quartiles?

Authors response: We appreciate the observation and we apologize because we had not updated this information.

We inform you that the classification of food groups (desirable and undesirable) was not considered in the final analysis. We only consider the score (quartiles) described by the reviewer (number of food groups x number of days they were consumed). Therefore, we have deleted the text associated with desirable and undesirable foods.

Comments to the Author, 2. I would like to see the findings regarding dietary diversity as a protective factor against a diabetes diagnoses reported in the results section and commented on the discussion. It seems to me this is an important result with implications for public health campaigns promoting dietary improvements. It is also interesting that the pattern of results for this factor is different for men and women, with a significant association for women, but not for men.

Authors’ response: To answer this comment we now include the following paragraphs in results and discussion:

Results : “When we compare the diversity of consumption of food groups or DD, we observe that in the total population and women with the highest quintile of DD (fourth quintile) the OR of having diabetes was lower (total population 0.5 CI 95% 0.3-0.7; women 0.4 CI 95% 0.2-0.7) than in the first quintile (OR = 1.0)”.

Discussión: “Dietary management is important to prevent diabetes [30] and dietary diversity is inversely associated with the risk of developing diabetes. [31] In our analysis we found that in women and total population having a greater DD was associated with a lower probability ratio of being diagnosed with diabetes. In men we do not find that DD is associated with a lower risk of diabetes possibly because in some subpopulations such as Hispanics [32] the results are still inconsistent and it is necessary to use a methodology that measures DD more accurately.These findings should motivate the generation of new studies that analyze this association longitudinally to confirm the direction and magnitude of causality. If this association is confirmed, it would be advisable to design communication programs to promote DD as another strategy to prevent diabetes”.

Reviewer´s comment, 3. On Table 1, the estimates for individuals without kidney failure are missing.

Authors’ response: We appreciate the observation. We have added the missing values on Table 1.

Reviewer´s comment 4. On line 195, the “six” years appears incorrect. It should be 10 (if the reference point is 2006) or 4 (if the reference point is 2012).

Authors’ response: The authors have attended this comment. Now the text describes that it was four years.

Reviewer´s comment 5. On line 227, there seems to be a typo. I think the authors mean “…with hypertension who are unaware”.

Authors’ response: We have modified the text and now the following is included: "... with hypertension who are unaware... “

Reviewer´s comment 6. The sentence on lines 259-261 needs to be revised to make sense.

Authors’ response: The authors have attended this comment. Now we describe the following: “One of the strengths of the study is that the results are representative of the Mexican adult population and are the most recent data on the prevalence of diabetes. This information will help the decision makers in health policies to know the magnitude of this disease, main associated risk factors and diabetes control practices.”

---

## [Decision Letter · Decision Letter 2]

9 Mar 2020

Prevalence of previously diagnosed diabetes and glycemic control strategies in Mexican adults: ENSANUT-2016

PONE-D-19-29166R2

Dear Dr. Ramirez-Villalobos,

We are pleased to inform you that your manuscript has been judged scientifically suitable for publication and will be formally accepted for publication once it complies with all outstanding technical requirements.

With kind regards,

Naeti Suksomboon

Academic Editor

PLOS ONE

Additional Editor Comments (optional):

Reviewers' comments:

Reviewer's Responses to Questions

**Comments to the Author**

1. If the authors have adequately addressed your comments raised in a previous round of review and you feel that this manuscript is now acceptable for publication, you may indicate that here to bypass the “Comments to the Author” section, enter your conflict of interest statement in the “Confidential to Editor” section, and submit your "Accept" recommendation.

Reviewer #1: All comments have been addressed

2. Is the manuscript technically sound, and do the data support the conclusions?

Reviewer #1: Yes

3. Has the statistical analysis been performed appropriately and rigorously? 

Reviewer #1: Yes

4. Have the authors made all data underlying the findings in their manuscript fully available?

Reviewer #1: (No Response)

5. Is the manuscript presented in an intelligible fashion and written in standard English?

Reviewer #1: Yes

6. Review Comments to the Author

Reviewer #1: I have no further comments. All of my concerns have been adequately addressed in this revised version.

7. PLOS authors have the option to publish the peer review history of their article (what does this mean?). If published, this will include your full peer review and any attached files.

Reviewer #1: No

---

## [Editor Report · Acceptance letter]

24 Mar 2020

PONE-D-19-29166R2 

Prevalence of previously diagnosed diabetes and glycemic control strategies in Mexican adults: ENSANUT-2016 

Dear Dr. Ramirez-Villalobos:

I am pleased to inform you that your manuscript has been deemed suitable for publication in PLOS ONE. Congratulations! Your manuscript is now with our production department. 

With kind regards,

on behalf of

Dr. Naeti Suksomboon 

Academic Editor

PLOS ONE